# ChromaFormer: A Scalable Multi-Spectral Transformer for Large-Scale Land Cover Classification

**Mingshi Li**  *mingshi.li@kuleuven.be*
*ESAT-PSI, KU Leuven*

**Dusan Grujicic**  *dusan.grujicic@esat.kuleuven.be*
*ESAT-PSI, KU Leuven*

**Stien Heremans**  *stien.heremans@inbo.be*
*INBO (Instituut Natuur- en Bosonderzoek)*

**Ben Somers**  *ben.somers@kuleuven.be*
*Department of Earth and Environmental Sciences, KU Leuven*

**Steven De Saeger**  *steven.desaeger@inbo.be*
*INBO (Instituut Natuur- en Bosonderzoek)*

**Matthew B. Blaschko**  *matthew.blaschko@kuleuven.be*
*ESAT-PSI, KU Leuven*

**Reviewed on OpenReview:** *https://openreview.net/forum?id=qzJVTJYEBc*

## Abstract

Remote sensing satellites such as Sentinel-2 provide high-resolution, multi-spectral imagery that enables dense, large-scale land cover classification. However, most deep learning models used in this domain—frequently CNN-based architectures—are limited in their ability to process high-dimensional spectral data and scale with increasing dataset sizes. Moreover, while transformer architectures have recently been introduced for remote sensing tasks, their performance on large, densely labeled multi-spectral datasets remains underexplored.

In this paper, we present ChromaFormer, a scalable family of multi-spectral transformer models designed for large-scale land cover classification. We introduce a novel Spectral Dependency Module (SDM) that explicitly learns inter-band relationships through attention across spectral channels, enabling efficient spectral-spatial feature fusion. Our models are evaluated on the Biological Valuation Map (BVM) of Flanders, a large, densely labeled dataset spanning over 13,500 km² and 14 classes. ChromaFormer models achieve substantial accuracy gains over baselines: while a 23M-parameter UNet++ achieves less than 70% accuracy, a 655M-parameter ChromaFormer attains over 96% accuracy. We also analyze performance scaling trends and demonstrate generalization to standard benchmarks. Our results highlight the effectiveness of combining scalable transformer architectures with explicit spectral modeling for next-generation remote sensing tasks.

## 1 Introduction and Background

Remote sensing imagery provides crucial information for applications in environmental monitoring, urban planning, and disaster management, yet the high dimensionality and volume of satellite data pose significant challenges for traditional analysis techniques. Deep learning models, particularly CNNs, have greatly advanced remote sensing analysis (Maggiori et al., 2016; Hu et al., 2015; Zhu et al., 2017; Li et al., 2022; Roy et al., 2020), but challenges remain in scalability and spectral feature learning. Vision transformers are

recently proven to be more effective on capturing long-range dependencies in remote sensing applications (Dosovitskiy et al., 2021). Among many transformer variations, the Swin Transformer further improved efficiency via localized self-attention windows, and subsequent studies applied transformers to multi-modal remote sensing imagery with great success (Aleissaee et al., 2023; Roy et al., 2023; Bergamasco et al., 2023), indicating careful architectural choices may be crucial for extracting meaningful features from multi/hyper-spectral data. Additionally, recent hybrid models (Yuan et al., 2023; Zhang et al., 2023) and large-scale pretraining techniques (Wang et al., 2024; 2022b) further showcase the direction of scaling vision models in remote sensing.

## 1.1 Scaling challenges in remote sensing

Despite the progress in model design, the scaling properties of neural networks for remote sensing remain under-explored. Intuitively, larger models and larger training datasets can yield better performance, as observed in computer vision and NLP domains (Brown et al., 2020). However, most remote sensing models to date are relatively small (often <200 million parameters) and are evaluated on limited datasets. For example, it is common to apply a 25M-parameter ResNet50 model to both a tiny hyperspectral scene (Firat & Hanbay, 2021) and a much larger satellite image collection (Papoutsis et al., 2021). Such a one-size-fits-all approach ignores potential efficiency and accuracy gains when matching model capacity to dataset scale. With the rise of large, labeled datasets (e.g. province/nation-scale), it becomes crucial to investigate how increasing model size and complexity impacts performance. Models such as scaled Swin-transformer variants (Peng et al., 2023; Yuan et al., 2023) have demonstrated the capability of scalable remote sensing architectures. However, it must be tested on a realistic dense dataset. For this study we use the Biological Valuation Map (BVM) of Flanders, a land-cover dataset with complete annotations for an area of over 13,500 km² (De Saeger et al., 2017; De Saeger et al., 2020; Li et al., 2024). A comparison of dataset sizes and corresponding model complexities is shown in Figure 1 and Table 2 (Appendix).

## 1.2 Leveraging spectral information

Another limitation of many existing models is the under-utilization of rich spectral information. Most CNN-based frameworks were originally designed for RGB imagery and struggle to utilize the spectral information provided by earth observation satellites such as Sentinel (VITO, 2020) and Landsat (Earth Resources Observation and Science (EROS) Center, 2020). Prior works have proposed spectral attention modules to enhance CNNs' performance (Hang et al., 2021; Roy et al., 2021; 2020). A transformer-based spectral–spatial network was even explored via neural architecture search (Zhong et al., 2022). Recent work such as ShapeFormer (Lv et al., 2023), AerialFormer (Hanyu et al., 2024), and LSKNet (Li et al., 2025) also demonstrated improvements in remote sensing classification by combining multiscale spatial and spectral modeling. Furthermore, self-supervised models like SSL4EO (Wang et al., 2023) show promise in pretraining with rich spectral bands. However, these methods typically model spectral correlations in a limited or sparse manner (e.g. separate 3D convolution blocks for spectral features) rather than a unified token-level attention across all bands. In other words, existing approaches do not fully learn inter-band relationships in an end-to-end fashion, leaving an opportunity to design architectures that more directly attend to spectral dependencies.

In this work, we address these gaps by proposing ChromaFormer, a multi-spectral transformer architecture tailored for multi-scale land cover segmentation. It is built on a Swin Transformer backbone and augmented with a novel Spectral Dependency Module that learns joint representations from all spectral bands. The selection of the Swin Transformer as a foundational architecture is a direct response to the dual challenges of global context modeling and computational scalability inherent to high-resolution remote sensing. While CNNs excel at extracting local features, they are constrained by a limited receptive field, failing to capture the long-range spatial dependencies crucial for complex land-use/land-classification tasks. Standard Vision Transformer (ViT) architectures, while capable of modeling global context , introduce a different prohibitive bottleneck: a computational complexity that scales quadratically with the input image size. This quadratic cost makes naive ViT models impractical for dense, pixel-level prediction on high-resolution satellite imagery. By integrating spectral attention into the transformer's self-attention layers, our model can weight and fuse information across different spectra – unlike prior CNN-based spectral modules that operate on fixed or handcrafted spectral groupings. We systematically study how the performance of ChromaFormer and other

architectures varies with model size and training data size, leveraging a large scale Sentinel-2 dataset (Li et al., 2024).

### 1.3 Experimental Testbed: The BVM Dataset

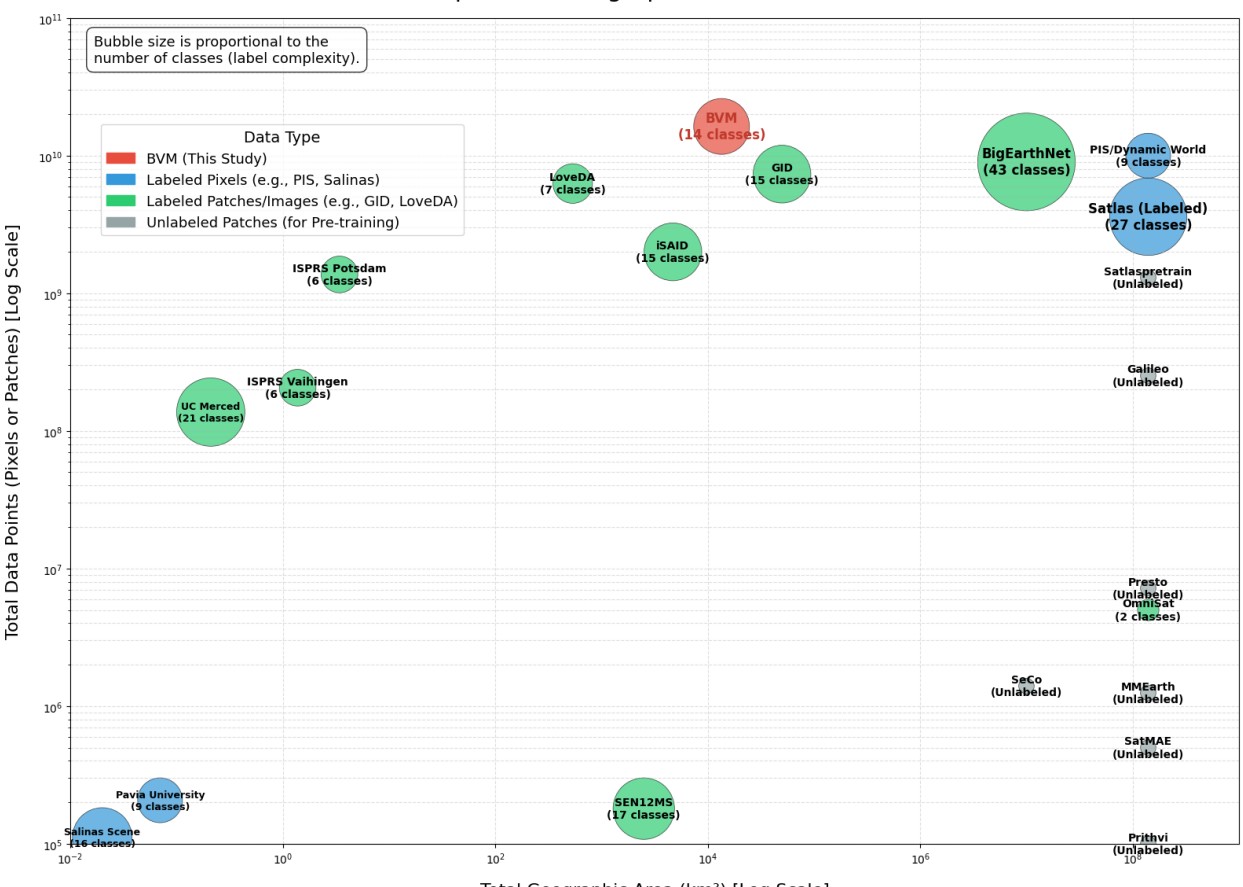

Figure 1: Dataset Comparison: Geographic Area vs. Total Data Points. Bubble size is proportional to the number of classes (label complexity). Datasets are sourced from BVM (Li et al., 2024) or their respective publications. Labeled pixel datasets include PIS/Dynamic World, Satlas, Pavia University, and Salinas Scene (Brown et al., 2022; Bastani et al., 2023; Gamba, 2003; Center for Hyperspectral Remote Sensing Scenes (EHU/UPV), 1998). Labeled patch/image datasets include BigEarthNet, GID, LoveDA, iSAID, ISPRS, UC Merced, OmniSat, and SEN12MS (Sumbul et al., 2019; Tong et al., 2020; Wang et al., 2021; Zamir et al., 2019; ISPRS Commission II / WG4, 2012a;b; Yang & Newsam, 2010; Astruc et al., 2024; Schmitt et al., 2019). Unlabeled pre-training datasets are also shown (Bastani et al., 2023; Ulbricht et al., 2025; Tseng et al., 2023; Manas et al., 2021; Nedungadi et al., 2024; Cong & Zhou, 2022; Jakubik et al., 2023).

To meaningfully evaluate the scaling properties of the chosen architectures, a correspondingly large-scale and high-quality dataset is required. The Biological Valuation Map (BVM) (Li et al., 2024) of Flanders was selected as it is positioned at the intersection of three critical criteria: scale, density, and quality. We argue that the BVM dataset is suited and necessary for meaningfully evaluating the scaling properties of large-capacity remote sensing models.

Large Scale: Large-capacity models, such as Swin-h and ChromaFormer-h, are data-hungry. They require a massive dataset to "unlock their potential and avoid overfitting". The BVM provides this necessary scale,

covering over 13,500 km². This is over 25 times larger than other common densely labeled benchmarks like LoveDA and is one to two orders of magnitude larger (in number of labeled pixels) than classic benchmarks. This scale permits a true big-data, big-model analysis.

High Density: The BVM provides dense, pixel-wise ground truth for the entire region , making it a valid and information rich testbed. This full-coverage pixel-level density is a key feature. In contrast, other large remote sensing datasets, such as BigEarthNet or LoveDA, are noted to have labels that are often sparse or less reliable.

High-Fidelity Quality: The BVM's label quality is a critical differentiator. It is not automatically generated, which can suffer from misinformation. Instead, it is a field-driven survey curated by vegetation experts (De Saeger et al., 2017; De Saeger et al., 2020). This expert-derived, high-fidelity ground truth ensures that the high accuracy scores achieved in the study are a robust measure of model capability, not an artifact of label noise.

In summary, this unique combination of massive scale, pixel level density, and expert verified quality makes the BVM an ideal and necessary benchmark for evaluating large-scale remote sensing models. In this work, it is paired with Sentinel-2 imagery to evaluate land cover classification models. Sentinel-2, operated by the European Space Agency (ESA), is a multispectral satellite system providing 13 spectral bands across visible, near-infrared, and shortwave infrared regions at resolutions of 10–60m. Similar to Landsat, Sentinel-2 provides higher temporal resolution and richer spectral information. Li et al. (2024) partitioned the BVM dataset into training and evaluation sets with nearly identical class distributions, ensuring that model performance is assessed on representative data. The Chi-squared distance between data splits was reported. Chi-squared distance is often used to determine the similarity of two categorical distributions, and it is formulated as follows: $D_{\chi^2}(P, Q) = \sum_i \frac{(P(i)-Q(i))^2}{P(i)+Q(i)}$, where P(i) and Q(i) are the probabilities of the i-th element in distributions P and Q, respectively, as per Table 3 in the supplementary material.

To complement our evaluation, we also consider Pavia University (Gamba, 2003) and Indian Pines (Baumgardner et al., 2015) benchmarks. These two datasets are widely used in hyperspectral classification, especially in evaluating channel-wise modeling performance. Prior benchmarks on these datasets include CNN-based architectures (Li et al., 2022), attention-guided fusion (Hang et al., 2021), token-mixing spectral transformers (Jeong et al., 2025), and WaveMix (Jeevan & Sethi, 2024), which applies Fourier/wavelet decomposition to image patches.

## 2 Methodologies

This section outlines the design and implementation of our proposed ChromaFormer framework on the BVM dataset. We first introduce the key component of ChromaFormer: the Spectral Dependency Module (SDM). We then emphasize ChromaFormer's architectural components, embedding and training strategies, which together enable effective modeling of both spectral dependencies and spatial structures inherent to multi/hyperspectral data. We also highlight how our design improves on scalability concerns (Hafner et al., 2024), that model performance may plateau or degrade if training regimes are not tuned to match the model depth and data volume.

### 2.1 Spectral Dependency Module (SDM)

To better utilize spectral information in multi-spectral imagery, we propose the Spectral Dependency Module (SDM), designed to explicitly model inter-band relationships.

Given an input tensor $X \in \mathbb{R}^{B \times C \times H \times W}$, we reshape the spatial dimensions to produce $Q, K, V \in \mathbb{R}^{B \times C \times N}$, where $N = H \times W$, and compute a spectral attention map of size $C \times C$. Then the channel-wise attention map is computed using scaled dot-product attention across spectral bands:

$$A = \text{softmax}\left(\frac{QK^{\top}}{\sqrt{N}}\right) \in \mathbb{R}^{B \times C \times C}.$$

The output is obtained by multiplying the attention weights with the value tensor:

$$O = A \cdot V \in \mathbb{R}^{B \times C \times N},$$

which is then reshaped back to $\mathbb{R}^{B \times C \times H \times W}$ to match the original feature map dimensions.

This mechanism enables the model to learn inter-band spectral dependencies across the full spatial extent of the image, thereby enhancing spectral-spatial feature representations. Unlike spatial attention, SDM emphasizes global interactions between spectral bands, and can be easily inserted into existing convolutional or transformer architectures due to its simple interface. It is worth mentioning that SDM is a lightweight, end-to-end learnable attention module that models channel-to-channel interactions, unlike fixed or non-learnable spectral priors. SDM is also resolution-independent and highly parallelizable, making it compatible with vision transformer pipelines.

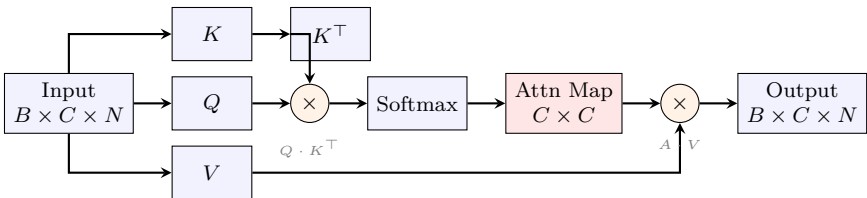

Figure 2: Architecture of the Spectral Dependency Module (SDM). The module computes a $C \times C$ attention map to capture global inter-band dependencies, where $C$ is embedded dimensions.

## 2.2 Relationships to prior works

While channel attention mechanisms such as Squeeze-and-Excitation (SE) Blocks (Hu et al., 2018) and the Convolutional Block Attention Module (CBAM) (Woo et al., 2018) have been adopted for spectral feature utilisation, the proposed Spectral Dependency Module (SDM) diverges from these in both objective and architecture.

Standard mechanisms typically employ a "Squeeze" operation—often a Global Average Pooling—to collapse spatial dimensions into a single descriptor per channel. This is followed by an "Excitation" step that produces a $C \times 1$ vector of scalar weights. SE-style modules perform independent channel re-weighting, where each band's contribution is scaled based on its global importance.

In contrast, SDM models pairwise dependencies between channels. Rather than generating a single weight per channel, the SDM computes a $C \times C$ spectral dependencies matrix. This allows the network to capture complex, non-linear correlations between distinct parts of the spectrum—interactions that are often ignored by scalar re-weighting. Furthermore, while traditional spectral indices such as NDVI rely on fixed, pre-defined ratios between specific bands, the SDM learns these relationships directly from the data. This learnable aspect is critical for identifying subtle spectral signatures necessary to distinguish classes with high intra-class variance, such as varying crop phenologies or mineral compositions.

## 2.3 Transformer backbone

Following SDM, we employ a hierarchical Swin Transformer backbone composed of four stages. Each stage consists of multiple Swin Transformer blocks with increasing receptive field and hidden dimensions. Our ChromaFormer models are defined by their specific configurations of embedding channels, the number of layers per stage (depths), and the attention head configurations across these stages. These parameters dictate the model's capacity and its ability to aggregate information across local and global spatial contexts, complementing the SDM's spectral modeling.

The configurations for the different ChromaFormer variants are as follows:

- **ChromaFormer-t (Tiny):** Uses 96 embedding channels, depths of $(2, 2, 6, 2)$ layers per stage, and attention head configurations of $(3, 6, 12, 24)$.

- **ChromaFormer-s (Small):** Similar to ChromaFormer-t in embedding channels (96) and attention heads $(3, 6, 12, 24)$, but with increased depths of $(2, 2, 18, 2)$ layers per stage.

- **ChromaFormer-b (Base):** Uses 128 embedding channels, depths of $(2, 2, 18, 2)$, and attention head configurations of $(4, 8, 16, 32)$.

- **ChromaFormer-l (Large):** Uses 192 embedding channels, depths of $(2, 2, 18, 2)$, and attention head configurations of $(6, 12, 24, 48)$.

- **ChromaFormer-h (Huge):** The largest variant, uses 352 embedding channels, depths of $(2, 2, 18, 2)$, and attention head configurations of $(11, 22, 44, 88)$.

The projected embedding length refers to the number of embedding channels ($C$) output by the Patch Embedding layer. For example, the ChromaFormer-t variant, the 13 input spectral channels are projected to a dimension of $C = 96$ using convolutional patch embedding. This 96-dimensional vector represents the initial spectral-spatial feature for each image patch. Similar to CNNs, the hierarchical nature of Swin-like transformers merges image patches at deeper layers. This effectively increases the receptive field, allowing the model to attend to larger spatial regions of the original image as the network goes deeper.

This systematic scaling of parameters allows us to investigate the performance and efficiency of ChromaFormer across a wide range of model capacities.

## 2.4 Training strategy and evaluation metrics

Models are trained with Adam optimizer on $4\times$NVIDIA A100-80G GPUs using DDP and HuggingFace Accelerate, we adopted distributed data parallelism with shared gradient synchronization across the GPUs. Batch size is 16, learning rate is 1e-4. Inputs and labels are normalized and padded. We use standard cross-entropy loss and follow recent best practices for fine-tuning.

We train the model with a learning rate of $10^{-4}$ and a batch size of 16. The training loop is implemented using the Accelerate library to support multi-GPU setups. Standard cross-entropy loss is used as the training objective. We normalize input tensors and apply padding where necessary to fit the fixed-resolution transformer architecture. In addition to classic griding and patching methods (Chen et al., 2014; Li et al., 2016), we follow recent recommendations for pretraining and fine-tuning from SSL4EO (Wang et al., 2023) and LSKNet (Li et al., 2025), using normalized input and label maps.

We report overall accuracy (OA) and per-class accuracy as our main evaluation metrics. In line with standard benchmarks, we evaluate the model on both accuracy and robustness against underrepresented classes (Wang et al., 2024). We also evaluate the scalability trends of our models following observations from the foundation model literature (Hafner et al., 2024; Brown et al., 2020), highlighting where models reach saturation and where additional parameter growth no longer yields benefits. All reported metrics were calculated on the test set. The validation set was used for hyperparameter tuning and early stopping.

# 3 Results and Discussion

## 3.1 Per-class performance

The empirical analysis of per-class accuracy curves in Figure 3 reveals that ChromaFormer models consistently outperform their Swin Transformer counterparts in land cover classification, demonstrating higher peak accuracies and faster convergence rates across most land-cover classes. This performance edge is expressed even more in classes that inherently rely on spectral information for accurate discrimination. For instance, in "Coastal dune habitats," where floristic diversity correlates with spectral heterogeneity, and "Cultivated land," where dynamic spectral signatures indicate plant health and growth stages over the year, ChromaFormer models show substantial gains. In "Inland marshes" and "Marine habitats," where mixed water–vegetation signals and water column effects challenge classification, ChromaFormer's superior spectral processing capabilities are also evident. The confusion matrix in Figure 4 further confirms these trends,

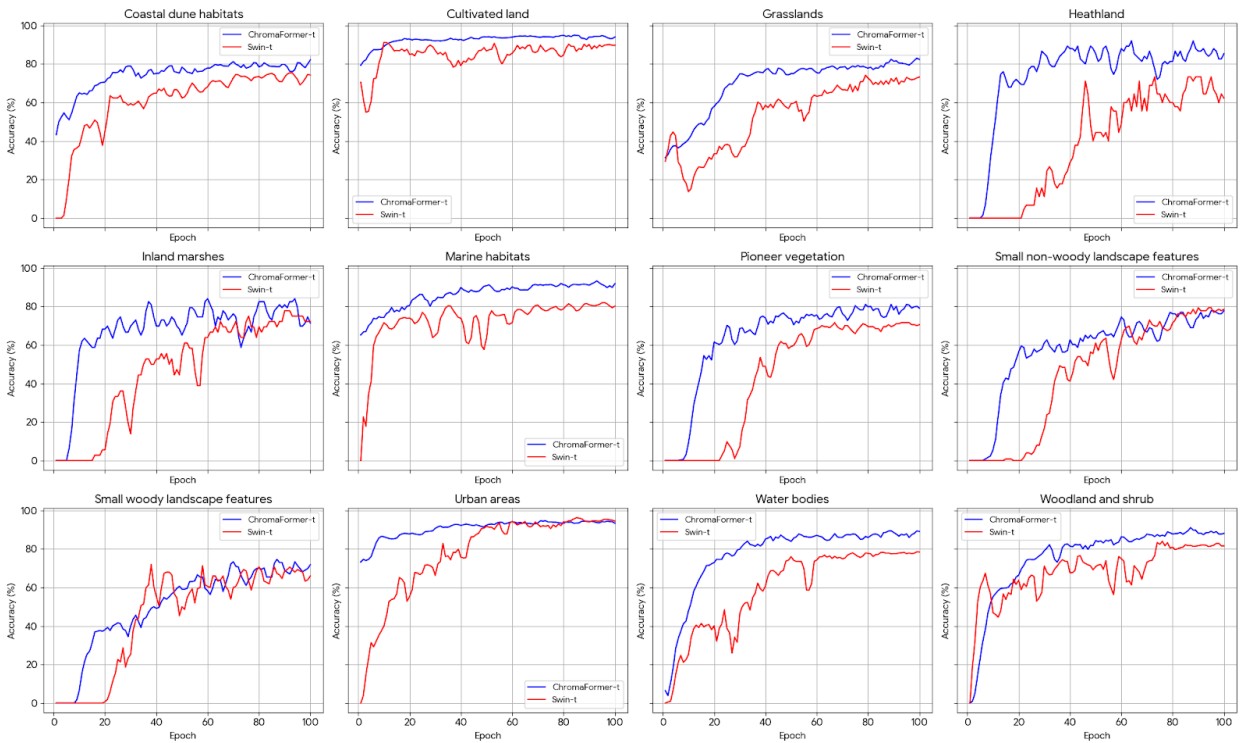

Figure 3: Per-class accuracy curves of Chromaformer-t (blue) and Swin-t (red), "Unknown" and "Small landscape features-not specified" removed as they provide less meaningful information.

showing strong diagonal dominance for spectrally complex classes, particularly in minority categories such as CDH, MH, and IM, where misclassification rates remain low despite severe class imbalance.

## 3.2 Quantitative model comparison

As shown in Table 1, the ChromaFormer models exhibit superior performance compared to both the ResNet family and the UNet++ model across all sizes, based on the provided data. In the small model category, ChromaFormer-t (27 million parameters) achieves an accuracy of 90.87%, significantly outperforming ResNet-20M (20 million parameters) with 80.95% accuracy and UNet++ (23 million parameters) with 68.81% accuracy. This substantial accuracy gap highlights the efficiency of ChromaFormer models in handling complex tasks, indicating that ChromaFormer could be more parameter-efficient than its competitors. Additionally, ChromaFormer models maintain higher scaling coefficients than ResNet models, indicating more efficient scaling as model size increases.

Between the Swin Transformer and ChromaFormer models, the ChromaFormer models offer additional advantages, making them a better choice for multi-spectral segmentation tasks. The integration of the Spectral Dependency Module (SDM) into the Swin Transformer architecture allows ChromaFormer models to capture spectral dependencies more effectively, leading to higher accuracies without a significant increase in parameters or loss of scaling efficiency. For example, ChromaFormer consistently achieves higher accuracy than Swin Transformers in almost all model size ranges, with almost identical scaling coefficients. This demonstrates that ChromaFormer models enhance spectral feature learning while maintaining efficient scalability. Therefore, the ChromaFormer models are optimal for multi-spectral tasks running at different scales, as they provide superior accuracy and maintain scaling efficiency compared to both their Swin Transformer counterparts and conventional models like ResNet and UNet++. Figure 5 shows output results from different

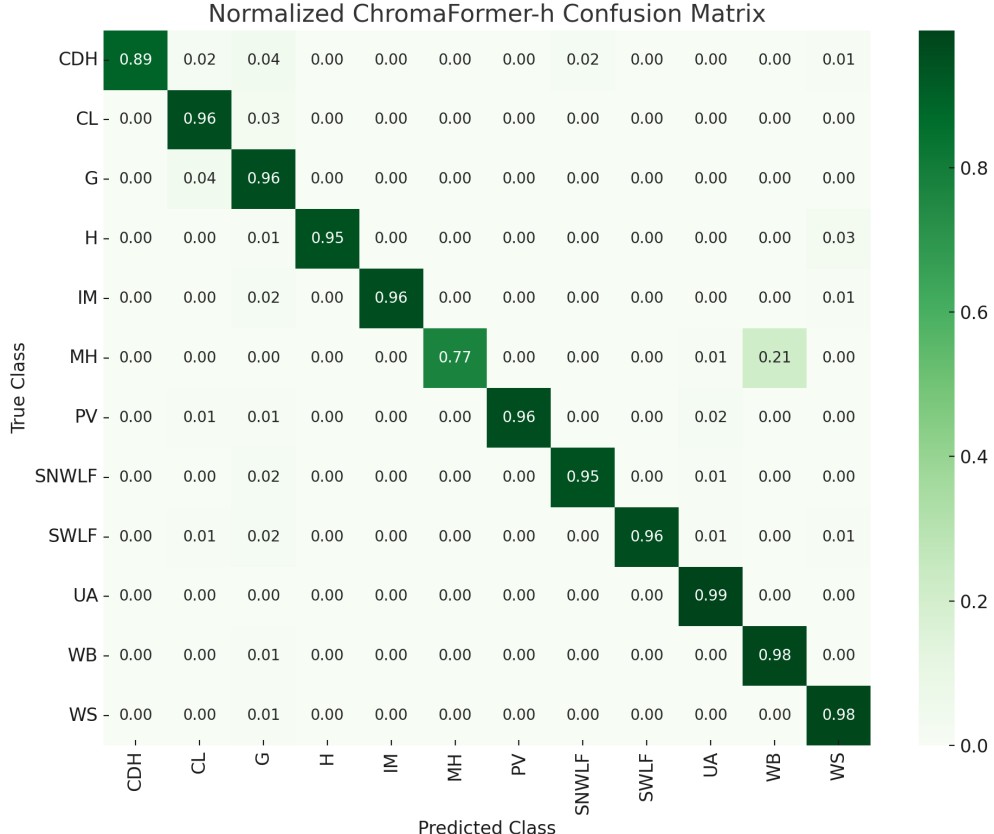

Figure 4: Normalized ChromaFormer-h confusion matrix on the test dataset. CDH = Coastal dune habitats, CL = Cultivated land, G = Grasslands, H = Heathland, IM = Inland marshes, MH = Marine habitats, PV = Pioneer vegetation, SNWLF = Small non-woody landscape features, SWLF = Small woody landscape features, UA = Urban areas, WB = Water bodies, WS = Woodland and shrub.

models; as we can see, ChromaFormer is better than ResNet and Swin Transformers in predicting minor classes.

### 3.3 Accuracy and loss analysis

Figure 6 compares the accuracy curves of ResNet-1M, ResNet-20M, ChromaFormer-t, Swin-t, and U-Net++ over 100 training epochs. ChromaFormer-t achieves the highest accuracy, steadily improving and reaching 90.87% by the end of training. Swin-t follows closely, reaching 88.98%. ResNet-20M converges and stabilizes earlier, indicating moderate capacity. ResNet-1M performs worse, plateauing at around 75.92%. U-Net++ shows the lowest performance, with accuracy saturating below 70% and minimal improvement after the initial epochs.

Figure 10 (Appendix) shows the loss curve comparison between ChromaFormer and ResNet. The analysis of loss curves for various neural network models reveals distinct behaviors based on model size and architecture. Smaller models like ResNet-1M and ResNet-20M show rapid early loss descent but quickly reach saturation, indicating a capacity limitation in handling complex patterns as training progresses. In contrast, larger ResNet models and all ChromaFormer models demonstrate a more gradual loss reduction, suggesting that they can continue improving with extended training due to their greater capacity to learn complex data representations.

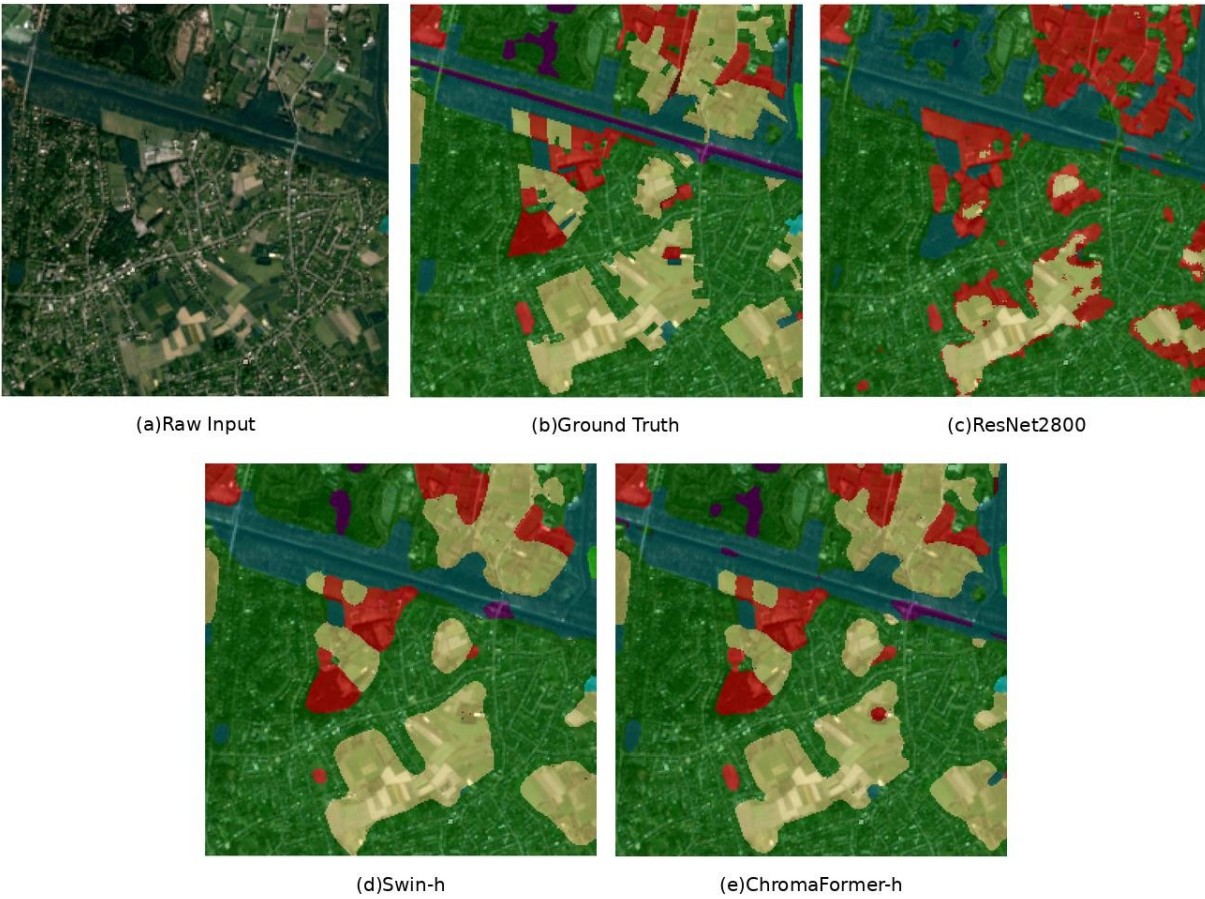

Figure 5: Qualitative results: (a) Raw Input, (b) Ground truth, (c) ResNet2800M, (d) Swin-h, and (e) ChromaFormer-h.

ChromaFormer models exhibit a slower but more persistent decline in loss, suggesting that they benefit from extended training epochs before showing signs of saturation, unlike conventional models like ResNet. This prolonged effectiveness in learning indicates that SDM modules are capable of exploiting their architectural efficiency to handle complex data relationships over longer periods. While larger ResNet models tend to plateau earlier, larger ChromaFormer models (like ChromaFormer-l and ChromaFormer-h) continue to show potential for improvement beyond the typical saturation points of conventional architectures, highlighting the distinct advantage of transformer-based models in sustained learning capability. We also observe that scaling beyond a certain point stops being rewarding for ResNet (e.g., the loss gap between ResNet-1550M and ResNet-2800M becomes negligible around $1 \times 10^8$ sample passes), whereas the losses for ChromaFormer-l and ChromaFormer-h only start to converge around $2 \times 10^8$ sample passes. This underscores the fact that transformer models, despite their size, are inherently designed to scale more effectively with increasing data and training duration before experiencing diminishing returns.

### 3.4 Scaling efficiency comparison

Figure 7 illustrates the relationship between model accuracy and scaling coefficient across a range of architectures, including ResNets, Swin Transformers, U-Net++, and ChromaFormers. While most models exhibit a general trend where higher scaling coefficients correlate with increased accuracy, ChromaFormer models consistently achieve superior accuracy even at moderate or low scaling coefficients. Notably, ChromaFormer-

Table 1: Comparison of models with parameters, training time, accuracy (with 95% error bars computed using $N = 21,500,000$ samples), and scaling coefficients. Scaling efficiency coefficient $S$ quantifies how effectively a neural network scales its performance relative to the increase in parameters and computational resources. It is mathematically defined as: $S = -\log\left(\frac{G}{P \times C}\right)^{-1}$ where: $G$ is the Performance Gain Factor, $P$ is the Parameter Count Scaling Factor, $C$ is the Computation Increase Factor.

| Model | Parameters (M) | Time/Epoch (h) | Accuracy (%) | Scaling Coefficient |
|---|---|---|---|---|
| Small Models (∼1M to 30M Parameters) | | | | |
| ResNet-1M | 1 | 0.8 | $75.92 \pm 0.02$ | Baseline |
| ResNet-2M | 2 | 1.0 | $76.03 \pm 0.02$ | 1.09 |
| UNet++ | 23 | 1.0 | $68.81 \pm 0.02$ | N/A |
| ResNet-20M | 20 | 1.0 | $80.95 \pm 0.02$ | 0.32 |
| Swin-t | 27 | 2.2 | $88.98 \pm 0.01$ | Baseline |
| ChromaFormer-t | 27 | 2.2 | $90.87 \pm 0.01$ | Baseline |
| Medium Models (∼50M to 100M Parameters) | | | | |
| ResNet-230M | 230 | 1.8 | $84.10 \pm 0.02$ | 0.16 |
| Swin-s | 49 | 3.1 | $92.19 \pm 0.01$ | 1.10 |
| ChromaFormer-s | 49 | 3.1 | $93.47 \pm 0.01$ | 1.09 |
| Swin-b | 86 | 3.7 | $93.08 \pm 0.01$ | 0.61 |
| ChromaFormer-b | 86 | 3.7 | $94.02 \pm 0.01$ | 0.61 |
| Large Models (∼150M to 300M Parameters) | | | | |
| ResNet-1550M | 1550 | 6.3 | $87.32 \pm 0.02$ | 0.11 |
| Swin-l | 195 | 4.7 | $94.57 \pm 0.01$ | 0.37 |
| ChromaFormer-l | 195 | 4.7 | $95.98 \pm 0.01$ | 0.37 |
| Extra-Large Models (∼650M to 2800M Parameters) | | | | |
| ResNet-2800M | 2800 | 10.0 | $89.19 \pm 0.01$ | 0.10 |
| Swin-h | 655 | 6.0 | $96.64 \pm 0.01$ | 0.24 |
| ChromaFormer-h | 656 | 6.0 | $96.67 \pm 0.01$ | 0.24 |

t, ChromaFormer-s, and ChromaFormer-b match or exceed the performance of their Swin counterparts at equivalent parameter scales. This suggests that ChromaFormer architectures offer improved scaling efficiency, achieving better accuracy without a proportional increase in computational cost.

### 3.5 Performance on benchmark datasets

To further validate the generalizability and effectiveness of our proposed ChromaFormer architecture, we conduct experiments on two widely used hyperspectral image classification benchmarks: Pavia University and Indian Pines. These datasets are standard testbeds for remote sensing models and have been extensively used to benchmark prior state-of-the-art methods.

Our lightweight variant, ChromaFormer-t, achieves a satisfying classification accuracy of 91.03% on the Pavia University dataset, it surpasses the 89% achieved by Hong et al. (2021). It also achives 99.70% on the Indian Pines dataset, surpassing the 92.37% achieved by autoencoder such as HyperspectralMAE (Jeong et al., 2025), 99.18% by multi-branch learning (Islam et al., 2024), and 98.45% by self-attention infused 3D-CNN (Reddy et al., 2024). Figure 8 clearly shows that ChromaFormer-t mis-classified only one pixel in the upper-left region, and for Figure 9, most pixels are classified correctly and class boundaries are well preserved. We emphasize that our model achieves these results without task-specific tuning or architectural overfitting to the dataset characteristics. The same model configuration was used across both datasets, underscoring its robustness in learning spatial–spectral representations. These results demonstrate that ChromaFormer is not only effective on a large-scale, complex dataset like BVM, but also achieves state-of-the-art performance on standard benchmarks, making it a compelling general-purpose solution for hyperspectral image classification tasks.

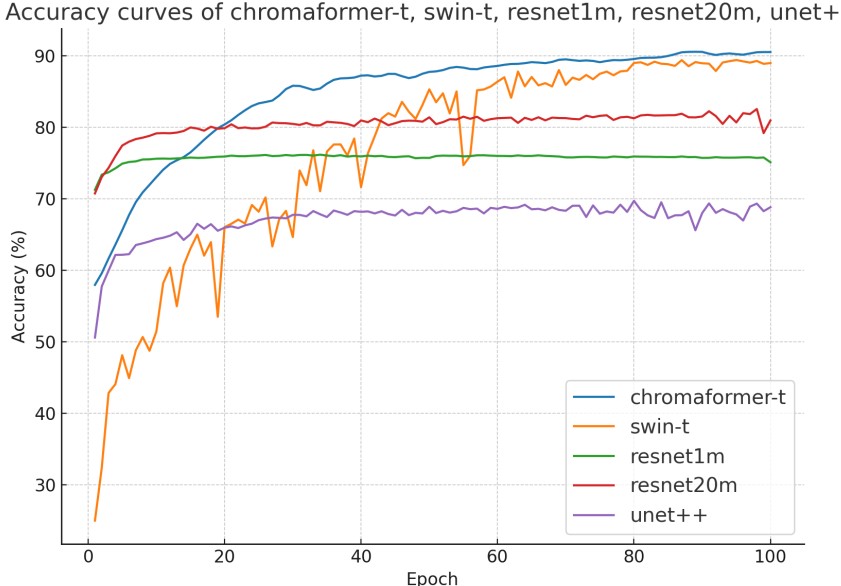

Figure 6: Accuracy curves of different models, "m" in legend stands for million parameters. We only plot small models for better visual clarity.

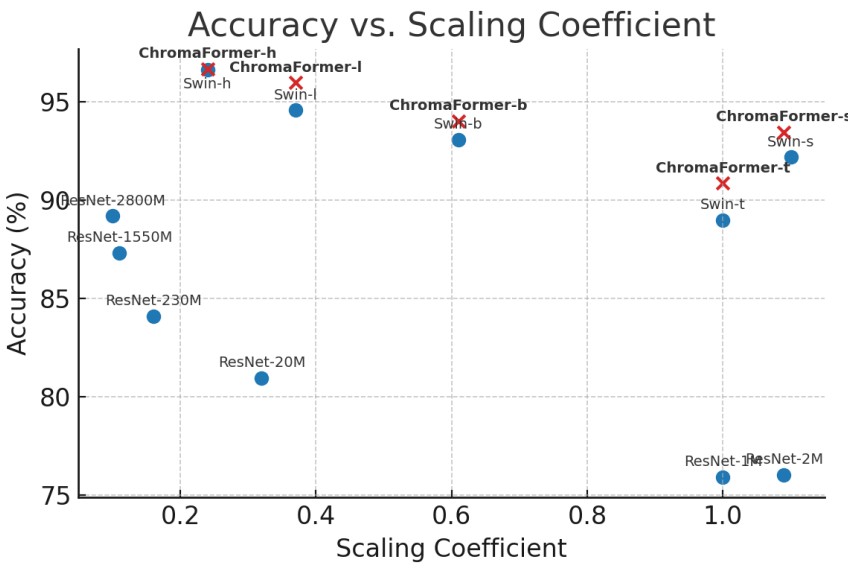

Figure 7: Accuracy versus scaling coefficient for various model architectures, including ResNet, Swin Transformer, U-Net++, and ChromaFormers. ChromaFormer models are highlighted with red crosses and bold labels. Notably, ChromaFormers achieve superior performance with equal or lower scaling cost compared to their counterparts.

## 4 Conclusions and limitations

In this work, through experiments on the large-scale Biological Valuation Map (BVM) of Flanders, Indian Pines and Pavia University dataset, we demonstrated that ChromaFormer outperforms conventional CNN-based models and pure vision transformers in terms of accuracy, scaling efficiency, and robustness to class imbalance. The novel SDM module improves spectral fusion in a lightweight, learnable manner. Our key

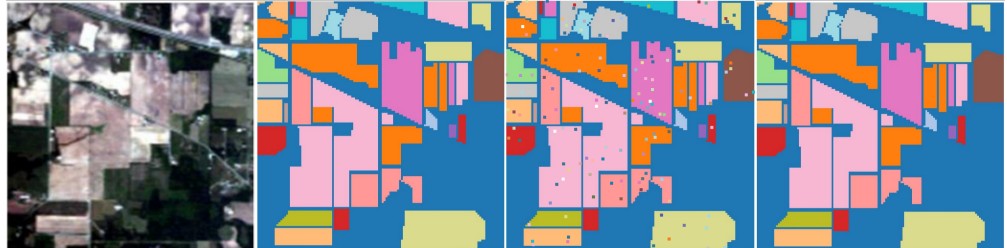

Figure 8: Visual comparison between ground truth and model prediction results on the Indian Pines dataset. (from Left to Right) Raw Input; Ground truth map with 16 semantic classes; Swin-t Output; Chromaformer-t Output generated using 5-fold cross-validated ensemble with majority voting; Prediction map overlaid with error regions: correctly predicted regions are shown with opacity.

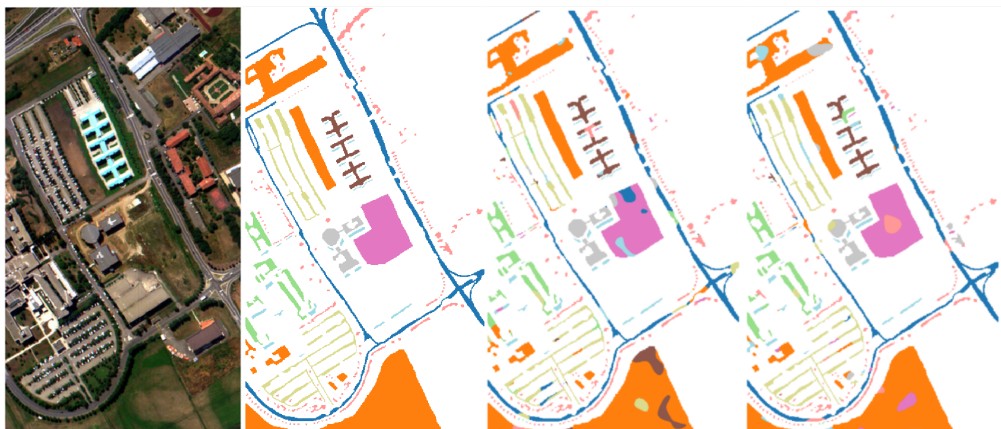

Figure 9: Visual comparison between ground truth and model prediction results on the Pavia University dataset. (from Left to Right) Raw Input; Ground Truth; Swin-t Output; Chromaformer-t Output; Outputs are generated using 5-fold cross-validated ensemble with majority voting.

finding is that aligning model complexity with dataset scale—both in terms of spatial coverage and spectral dimensionality—is crucial for effective land cover classification. The SDM module enables the model to leverage inter-band spectral correlations in a learnable, scalable manner, resulting in superior generalization and performance across different classes and resolutions. SDM's capability of utilizing temporal data is further detailed in Table 4 (Appendix). Additionally, ChromaFormer models maintain high scaling efficiency even at hundreds of millions of parameters, suggesting their suitability for processing increasingly large remote sensing datasets being released worldwide.

Nevertheless, our study has limitations. First, BVM dataset is geographically constrained to the Flemish region of Belgium, evaluation on more diversed geographic zones is necessary to validate global generalization. Second, computational constraints limited our ability to run extensive ablation tests with ChromaFormer-b/l/h variations and explore even larger model variants or longer training regimes, which could uncover further scaling benefits. Another limitation of this work is the limited ablation study of hyperparameters tuning of SDM block, one possible future direction could be finding a low-rank/sparse representation of spectral dependencies which saves computational power and consumption of memory. Finally, while we benchmarked against several popular baselines, additional comparisons with emerging foundation models or hybrid transformer–CNNs would help further position ChromaFormer within the broader model landscape. Future work may also investigate integrating SDM into other model families or extending the architecture to handle temporal sequences in multi-temporal satellite imagery.

**Acknowledgments**

This research received funding from the Flemish Government (AI Research Program) and the Research Foundation Flanders (FWO) through project number S006421N. We acknowledge the EuroHPC Joint Undertaking for awarding the project ID EHPC-BEN-2025B07-037, EHPC-BEN-2025B11-070 and EHPC-AIF-2025SC02-042 access to the EuroHPC supercomputer LEONARDO, hosted by CINECA (Italy) and the LEONARDO consortium. In addition, the resources and services used in this work were partially provided by the VSC (Flemish Supercomputer Center), funded by the Research Foundation- Flanders (FWO) and the Flemish Government.

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

## A    Supplementary material

In this appendix, we include Table 2, Table 4, Figure 10, and Figure 11, giving insights into comparative dataset sizes, multi-season composite analysis, model convergence, and SDM module placement ablation study, respectively.

The comparative data in Table 2 is compiled from research utilizing a variety of mainstream datasets and models. Key sources include work on Salinas Scene (Center for Hyperspectral Remote Sensing Scenes (EHU/UPV), 1998; Li et al., 2022; Liu et al., 2023; Roy et al., 2020; Liao et al., 2024), UC Merced (Yang & Newsam, 2010; Bi et al., 2022; 2021b;a; Özyurt et al., 2020), ISPRS Vaihingen and Potsdam (ISPRS Commission II / WG4, 2012b;a; Liu et al., 2022; Li et al., 2021a; 2023a; Chen et al., 2018; Liu et al., 2024; Li et al., 2023b; 2021b; Wang et al., 2022a; Hanyu et al., 2024), iSAID (Zamir et al., 2019; Guo et al., 2022), LoveDA (Wang et al., 2021; Dimitrovski et al., 2024; Hwang et al., 2024; Cha et al., 2024), GID (Tong et al., 2020; Li et al., 2025; Ovi et al., 2023), and BigEarthNet (Sumbul et al., 2019; Wang et al., 2023; 2022b), along with the BVM dataset used in this study (Li et al., 2024).

### A.1    Class Distribution across BVM Data Splits

As shown in Table  3, the class distribution of BVM dataset is listed.

### A.2    Seasonal Modeling Experiments

The improved performance demonstrated by the four-season composite models is grounded in the principle of phenology. Many land cover classes that are spectrally similar in a single season, such as "Cultivated land" and "Grasslands", exhibit unique temporal signatures throughout the year. Quantitatively, this is evidenced

Table 2: A comparison of mainstream datasets and the BVM dataset, and collection of models applied to them. Note that the pixel count (B: billion, M: million) is an estimation based on dataset specifications.

| Dataset | Pixels | Model | Accuracy |
|---|---|---|---|
| Salinas Scene | 0.11M | PSE-UNet | 91.01% (OA) |
| | | 3D-CNN | 97.55% (OA) |
| | | HybridSN | 99.84% (OA) |
| | | SMALE | 99.28% (OA) |
| UC Merced | 137M | DenseNet-121 | 99.88% (OA) |
| | | MS2AP | 99.01% (OA) |
| | | LSENet | 98.69% (OA) |
| | | VGG-VD16 | 95.21% (OA) |
| ISPRS Vaihingen | 206M | PGNet | 86.32% (OA) |
| | | MANet | 86.51% (OA) |
| | | EMNet | 95.42% (OA) |
| | | DeepLabv3+ | 86.07% (OA) |
| ISPRS Potsdam | 1.37B | CM-UNet | 91.86% (OA) |
| | | SSCNet | 91.03% (OA) |
| | | HCANet | 90.15% (OA) |
| | | AerialFormer-B | 91.4% (OA) |
| | | DC-Swin | 92% (OA) |
| iSAID | 2B | SegNeXt-L | 70.3% (IoU) |
| | | SegNeXt-B | 69.9% (IoU) |
| | | AerialFormer-B | 69.3% (IoU) |
| LoveDA | 6.27B | UNet-Ensemble | 56.16% (IoU) |
| | | SFA-Net | 54.9% (IoU) |
| | | ViT-G12X4 | 54.4% (IoU) |
| GID | 7.34B | LSKNet-S | 82.3% (OA) |
| | | DeepTriNet | 77% (OA) |
| BigEarthNet | 9B | ResNet50 | 91.8% (OA) |
| | | ViT-S | 89.9% (OA) |
| | | ResNet18 | 89.3% (OA) |
| **BVM** | **10.57B** | ChromaFormer-h | 96.67% (OA) |

by the 4-season composite boosting the ChromaFormer-s model to 93.47% OA, a significant +2.97% gain over even the strongest single season (Summer, 90.5%). This gain is also seen in the challenging "Cultivated Land" class, which jumps from 91.8% to 94.55% accuracy.

Summer season yields the best results among single-season analyses due to peak vegetation health and spectral separability, while Winter performs the poorest (e.g., 85.2% for ChromaFormer-s, over 5% lower than Summer), likely due to dormant vegetation and lower light conditions masking class distinctions.

Across all scenarios, the larger ChromaFormer-s model consistently outperforms the ChromaFormer-t model, maintaining a stable 2.5% OA advantage, which demonstrates the model's robust scalability. The Spectral Dependency Module (SDM) is particularly effective here. It treats the seasonal stack as a single high-dimensional spectral-temporal input, allowing it to explicitly learn the phenological curves by modeling the crucial inter-band relationships not just within a season, but across the entire temporal stack.

### A.3 Ablation on the placement of the Spectral Dependency Module (SDM)

To validate the architectural design of the ChromaFormer, we conducted an ablation study regarding the placement of the Spectral Dependency Module (SDM). Specifically, we compared our proposed early spectral fusion strategy against a late spectral fusion alternative.

Table 3: Class distribution across BVM data splits. (Li et al., 2024)

| Class | Train (%) | Val. (%) | Test (%) |
|---|---|---|---|
| Coastal dune habitats | 0.10 | 0.06 | 0.25 |
| Cultivated land | 34.27 | 33.92 | 32.85 |
| Grasslands | 23.07 | 22.92 | 22.14 |
| Heathland | 0.54 | 1.08 | 0.95 |
| Inland marshes | 0.24 | 0.23 | 0.22 |
| Marine habitats | 0.26 | 0.04 | 0.34 |
| Pioneer vegetation | 0.69 | 0.65 | 0.56 |
| Small landscape features - not specified | 0.05 | 0.06 | 0.07 |
| Small non-woody landscape features | 0.14 | 0.13 | 0.12 |
| Small woody landscape features | 0.63 | 0.58 | 0.73 |
| Unknown | 0.01 | 0.007 | 0.006 |
| Urban areas | 26.27 | 27.05 | 28.56 |
| Water bodies | 2.08 | 2.05 | 1.74 |
| Woodland and shrub | 11.65 | 11.22 | 11.47 |

Table 4: Performance of single-season vs. multi-season temporal composite models.

| Model Configuration | Overall Acc. (OA) | Cultivated Land Acc. | Woodland/Shrub Acc. |
|---|---|---|---|
| *ChromaFormer-t Models* | | | |
| Spring Season Only | 85.3% | 82.1% | 88.4% |
| Summer Season Only | 88.1% | 90.5% | 89.1% |
| Autumn Season Only | 86.5% | 84.3% | 87.2% |
| Winter Season Only | 82.7% | 79.8% | 83.5% |
| **4-Season Composite** | **90.87%** | **92.01%** | **91.53%** |
| *ChromaFormer-s Models* | | | |
| Spring Season Only | 88.0% | 85.9% | 89.5% |
| Summer Season Only | 90.5% | 91.8% | 91.3% |
| Autumn Season Only | 88.9% | 87.5% | 89.1% |
| Winter Season Only | 85.2% | 82.3% | 86.0% |
| **4-Season Composite** | **93.47%** | **94.55%** | **94.02%** |

Early Fusion: The SDM is inserted immediately after the patch embedding and before the first Transformer stage. This allows the model to recalibrate channel weights based on global spectral correlations before any spatial mixing occurs.

Late Fusion: The SDM is inserted within the Transformer stages, specifically after the Multi-Head Self-Attention (MSA) module in each block. This mimics the design of standard channel-attention mechanisms like SE-Blocks Hu et al. (2018) or CBAM Woo et al. (2018), where channel re-weighting is performed on spatially processed features.

As illustrated in Figure 11 the Early Fusion architecture consistently outperforms the Late Fusion variant in overall accuracy (OA). Specifically, we observe the following improvements when moving from Late Fusion to Early Fusion:

- ChromaFormer-t: Accuracy increased from 88.76% to 90.87% (+2.11%).

- ChromaFormer-b: Accuracy increased from 93.38% to 94.02% (+0.64%).

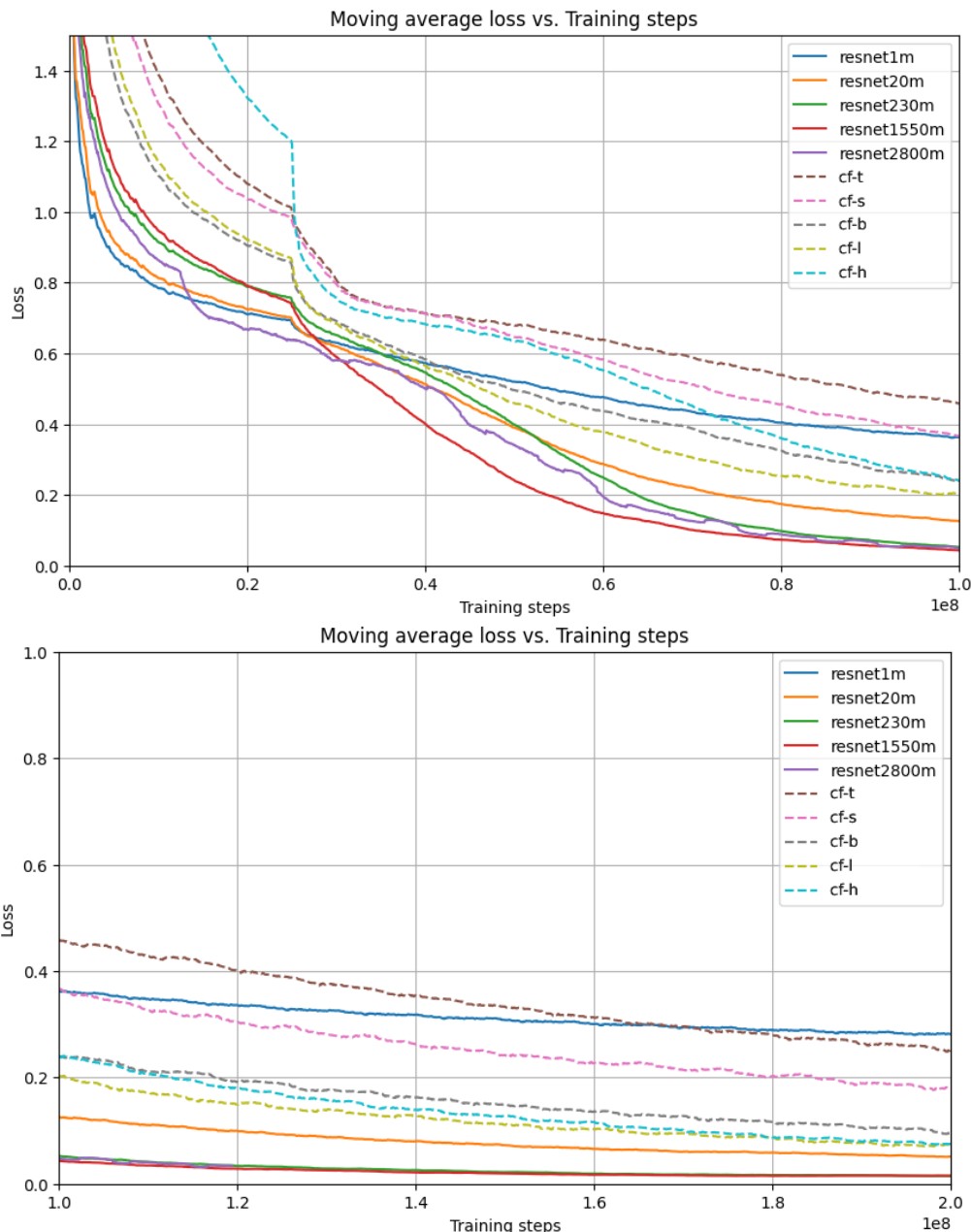

Figure 10: Loss curves of scaled networks, up: initial region of training, bottom: stable descending region. "cf" stands for ChromaFormer.

- ChromaFormer-l: Accuracy increased from 94.80% to 95.98% (+1.18%).

- ChromaFormer-h: Accuracy increased from 95.28% to 96.67% (+1.39%).

Most notably, the Early Fusion architecture resolves the performance bottleneck seen in the largest models. While the Late Fusion ChromaFormer-h (95.28%) lagged significantly behind the baseline Swin-h (96.64%), the proposed Early Fusion ChromaFormer-h (96.67%) successfully surpasses the Swin-h baseline. This suggests that as model capacity increases, decoupling spectral correlation (via Early Fusion) from spatial mixing becomes increasingly critical for maximizing accuracy.

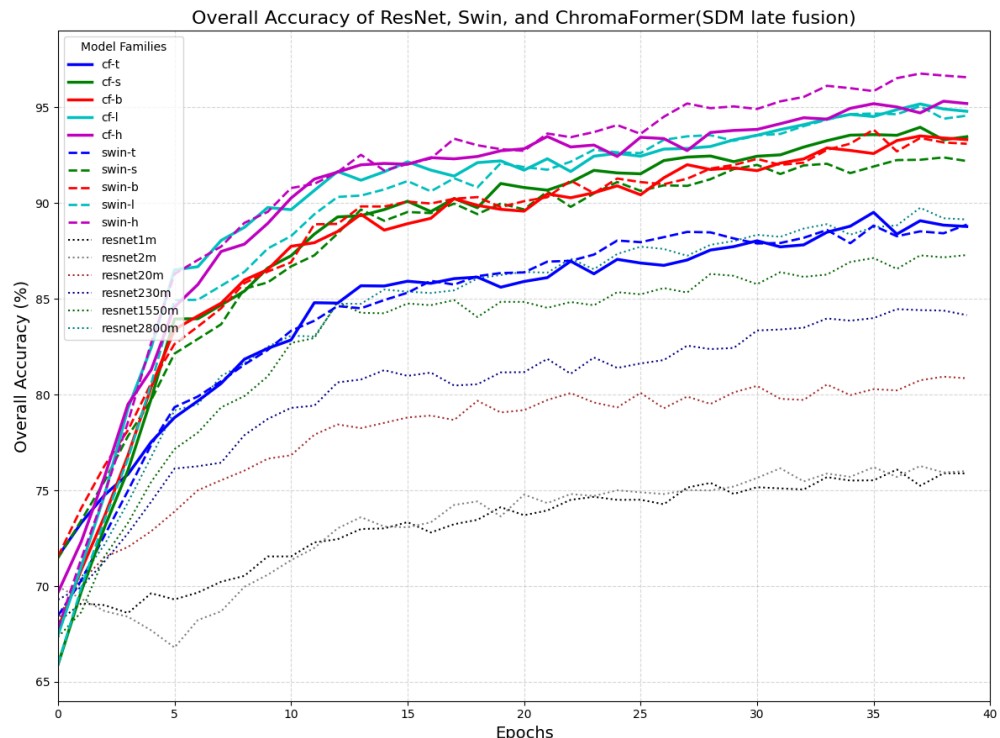

Figure 11: The overall accuracy curves of tested models, ChromaFormers in this figure are late fusion version

## A.4 Generalization Analysis on LandCoverNet(North America)

Table 5: Semantic aggregation used to align classes of the Biological Valuation Map (BVM) with classes of LandCoverNet North America (LCN-NA).

| Target Class (LCN-NA) | Source Classes (BVM) |
|---|---|
| Water | Water bodies |
| Artificial Bare Ground | Urban areas |
| Woody Vegetation | Woodland and shrub; Small woody landscape features |
| Cultivated Vegetation | Cultivated land |
| (Semi) Natural Vegetation | Grasslands; Heathland; Inland marshes; Pioneer vegetation; Coastal dune habitats; Marine habitats; Small non-woody landscape features |

We also extended our analysis by training chromaformer-t on the LandCoverNet(North America) dataset. Transitioning from the regionally homogeneous BVM dataset (Flanders) to a continental-scale dataset (North America) introduced significant challenges, particularly in biome-dependent classes.

For comparison, we merged some classes in BVM into classes similar to LCN-NA 5 12.While the model achieved excellent performance on physically distinct classes like Water (97.5%) and Woody Vegetation (91.5%) which could be benefiting from the cleaner labels of LandCoverNet, we observed a performance drop in Cultivated Vegetation (79.0%) and Semi-Natural Vegetation (74.0%). This decrease is likely due to the extreme intra-class variance of North American geography (only 7 classes in total).

Despite these continental-scale challenges, the model achieved a robust Overall Accuracy of 86.8%. This result confirms that while geographic heterogeneity impacts specific vegetation classes, the architecture effectively learns generalizable spectral features across diverse domains.

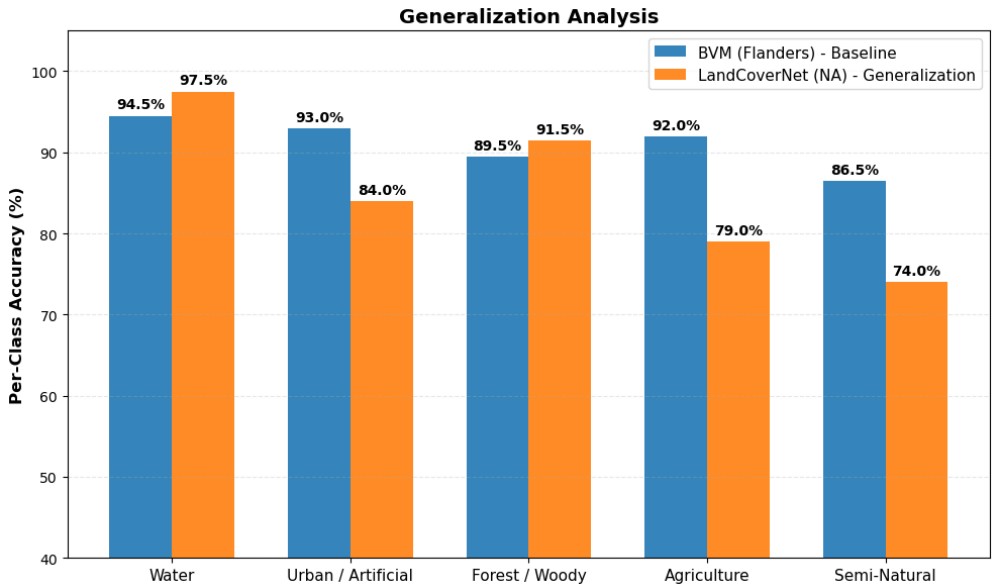

Figure 12: Per-class accuracy comparison on chromaformer-t between LandCoverNet (North America) and BVM.

## A.5 Comparison with prior channel attention modules

To validate the effectiveness of the Spectral Dependency Module (SDM), we benchmarked it against standard channel attention mechanisms: Squeeze-and-Excitation (SE) (Hu et al., 2018) and CBAM (Woo et al., 2018) by integrating them into the Swin-t backbone.

Table 6: Impact of different spectral/channel attention mechanisms on the Swin-t architecture (BVM Dataset).

| Model Variant | Global Pooling? | Acc. (%) | Accuracy Change |
|---|---|---|---|
| Swin-t (Baseline) | - | 88.49 | - |
| Swin-t + SE | Yes | 86.32 | -2.17 |
| Swin-t + CBAM | Yes | 87.11 | -1.38 |
| **ChromaFormer-t** | **No** | **90.62** | **+2.13** |

As shown in Table 6, switching from SDM to SE or CBAM degrades performance relative to the baseline, whereas SDM yields a +2.13% gain. We one possible reason could be the reliance of global average pooling technique used both in SE and CBAM. In dense land cover classification, this aggregates spatial features into a single vector, causing dominant classes to suppress the spectral features of smaller objects. In contrast, SDM models inter-band dependencies without spatial compression, preserving the local spectral context.

## A.6 Per-class performance comparison between Swin-b and Chromaformer-b

As shown in Figure 13, the proposed Chromaformer-b with considerably large parameter count and model size is consistently performing better than Swin-b for most classes ('small landscape features-not specified' is considered less meaningful due to very low amount of sample pixels.)

## A.7 Confusion matrix of Swin-t and Chromaformer-t on Pavia University Dataset

As shown in Figure 14, Chromaformer has a more stable performance across all land classes than its Swin counterpart.

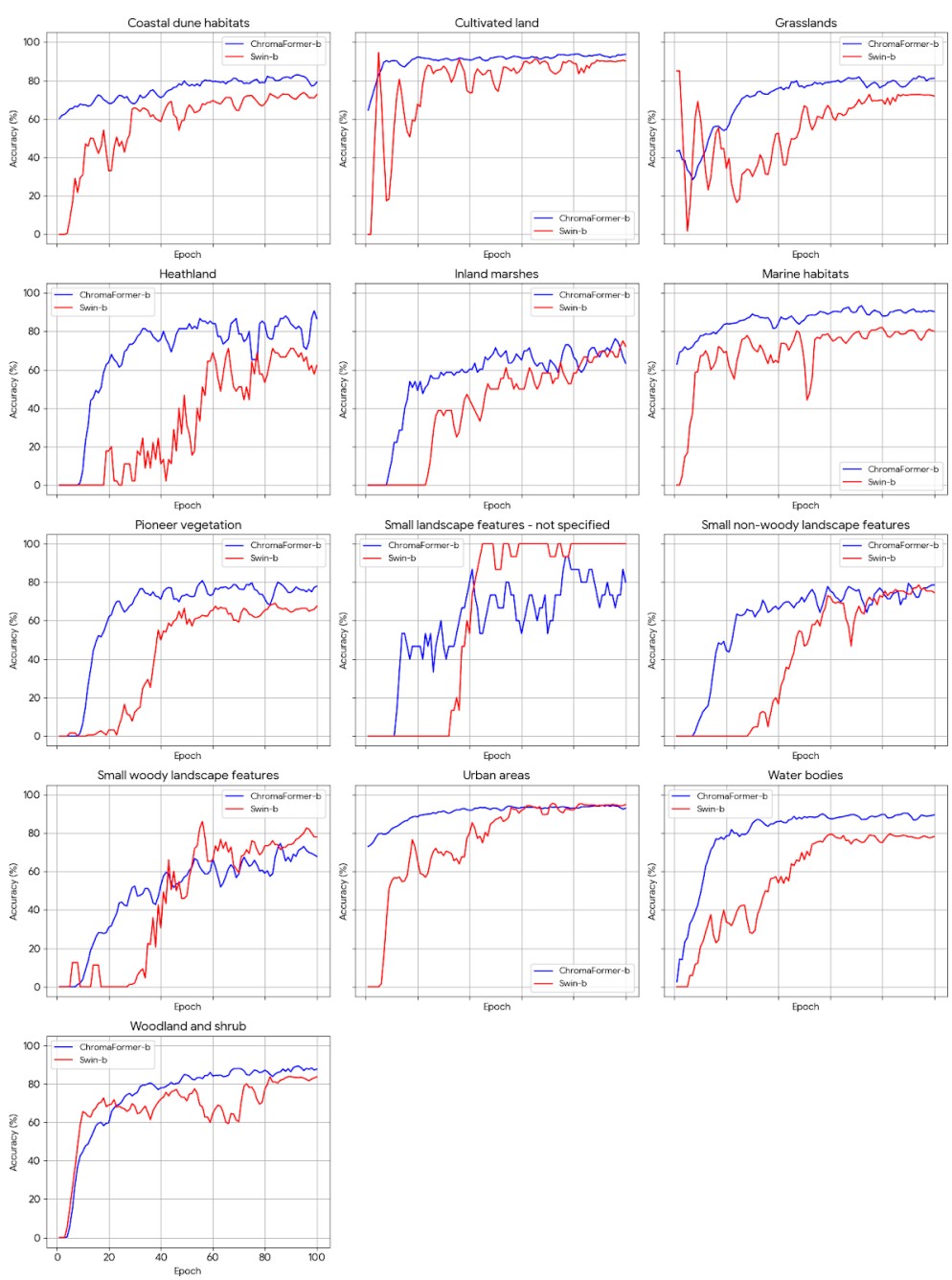

Figure 13: Per-class accuracy comparison between Swin-b and Chromaformer-b

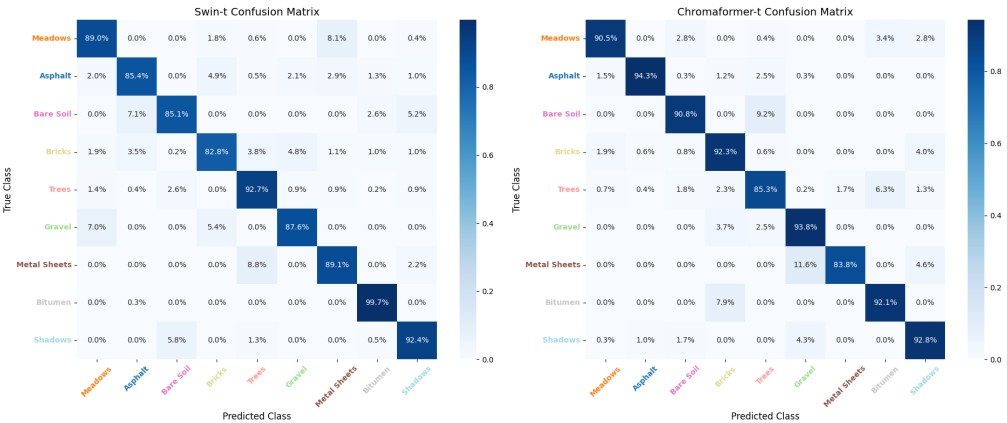

Figure 14: Confusion matrix calculated based on output of Swin-t and Chromaformer-t models

