# OpenReview forum: "ChromaFormer: A Scalable and Accurate Transformer Architecture for Land Cover Classification"
_TMLR — Accepted by TMLR_

### Review · Reviewer_YvKk · 2025-12-20

**Summary Of Contributions:**

The paper proposes a Spectral Density Module (SDM) for processing hyperspectral images in remote sensing. The experimental evaluation is primarily show quantitative results on a single dataset, namely the Biological Valuation Map (BVM). While the proposed approach shows promising results for this specific case, the generalization capability of the method remains unclear and insufficiently validated.

**Audience:**

No

**Audience Explanation:**

The proposed approach does not appear to be sufficiently novel, and the experimental results would benefit from deeper interpretation to better align with the interests of the TMLR audience.

**Broader Impact Concerns:**

There are no broader impact concerns as such.

**Claims And Evidence:**

No

**Claims Explanation:**

The claims made in the paper are supported by experimental results; however, the overall presentation lacks clarity. The proposed method would benefit from a more thorough and coherent description. Additionally, the experimental section should include further experiments to better support the main results and strengthen the credibility of the proposed approach.

**Requested Changes:**

Strengths:

- The paper evaluates the proposed approach across multiple baseline variants, including Chromaformer (tiny, base, large, and huge). The conclusion that Chromaformer-Huge outperforms other baselines and model variants appears reasonable and well supported by the reported results.


Weaknesses:

- The main methodological contribution is relatively limited. The proposed approach essentially adapts self-attention to operate across the channel (spectral band) dimension, learning relationships among spectral channels within a local patch. This idea, while useful, does not appear substantially novel.

- The paper lacks clarity and coherence in several sections and contains redundant content.
    - For instance, Section 1.3 is dedicated to the choice of dataset, which may give the impression that the dataset itself is a contribution of the paper. However, the dataset is only selected for experimentation, and this should be stated more clearly.
    - The detailed dataset analysis presented in Table 1 is not central to the paper’s contribution. This information could either be moved to the supplementary material or replaced with a citation to the original dataset paper.

- The overall presentation of the paper can be improved:
    - Figures 4 and 5 contain excessive white space. In particular, Figure 5 could be compressed into a single row, allowing room for additional examples that better illustrate the advantages of the proposed method over the baselines.
    - Similarly, Figures 6 and 7 could be combined into a single row for improved compactness and readability.

- Figure 2, which is the main architectural figure, only illustrates the attention component. While it is clear that self-attention is applied across channels, the figure does not sufficiently connect this component back to the full Chromaformer (Swin-based hierarchical transformer) architecture. This is something author should consider including.

- Section 2.2 would benefit from a more detailed explanation, along with an accompanying figure. This figure should ideally serve as the main architecture diagram, clearly showing how the proposed SDM integrates with the overall model.

- Figure 3 compares Swin-T and Chromaformer-T on the 12-class HSV dataset. Several questions arise:
    - Reporting overall accuracy alone would be sufficient, as the convergence trend discussed in the paper already indicates that Chromaformer achieves faster convergence.
    - Only the tiny variants are compared, despite the fact that the best performance is reported for the huge variants. Comparisons across larger architectures would strengthen the analysis.

- Figures 5, 8, and 9 present qualitative results using the proposed Chromaformer method but do not include the original input images. Including the original images would help readers understand where and why the method struggles.
    - Additionally, there is no state-of-the-art (SOTA) qualitative comparison for Figures 8 and 9 on the Indian Pines and Pavia University datasets.
    - In Figure 8, the prediction maps with error overlays are not clearly explained. It is unclear what the overlays represent and what aspects the reader should focus on. Moreover, Figures 8 and 9 are not adequately discussed in the main text.
    - There is also no quantitative SOTA comparison for the Indian Pines and Pavia University datasets, which would be expected given their widespread use.

- The paper places disproportionate emphasis on the BVM dataset. Detailed quantitative comparisons and a confusion matrix are provided only for BVM, while similar analyses are missing for the other two datasets.

- An ablation study of the SDM module is missing. For example, analyzing the impact of different embedding dimensions, number of attention heads, or other SDM-specific hyperparameters would strengthen the technical contribution.

---

> ### Author Response · Authors · 2026-02-19
>
> Dear reviewer,
>
> Thanks for your time and effort, we carefully read through your comments and made changes to the manuscript. Below are answers to your raised concerns. Main text related to these changes are marked in violet color.
>
> > The main methodological contribution is relatively limited....This idea, while useful, does not appear substantially novel.
>
> Answer: Novelty is not a criterion for review in TMLR.  Please see https://jmlr.org/tmlr/acceptance-criteria.html for review criteria.
>
> >The paper lacks clarity and coherence in several sections and contains redundant content.
> >>   For instance, Section 1.3 is dedicated to the choice of dataset, ..., and this should be stated more clearly.
>
> Answer: Section 1.3 is about the selection of datasets.  The BVM dataset is cited and attributed to the original contributors, while Section 1.3 details why we have made this choice for primary evaluation.  There are few densely labeled datasets of high quality covering this large of a surface area.
>
> >>   The detailed dataset analysis presented in Table 1 is not central to the paper’s contribution.... replaced with a citation to the original dataset paper.
>
> Answer: Table 1 gives the class distributions to assist in interpreting Figure 3.  However, we agree that it is better placed in the appendix as it is not central to the contributions of this submission.  Thanks for the suggestion.  We have moved it.  In the new numbering it is Table 3.
>
> >The overall presentation of the paper can be improved:
> >>    Figures 4 and 5 contain excessive white space. In particular, .... of the proposed method over the baselines.
> >>    Similarly, Figures 6 and 7 could be combined into a single row for improved compactness and readability.
>
> Answer: We believe the formatting is not an impediment to understanding the content of the figures.
>
> > Figure 2, which is the main architectural figure, only illustrates the attention component. While it is clear that .... architecture. This is something author should consider including.
>
> Answer: Thanks for the suggestion.  We have updated Figure 2 to give a more detailed view of the operations in the Spectral Dependency Module.
>
> > Section 2.2 would benefit from a more detailed explanation, .... the proposed SDM integrates with the overall model.
>
> Answer: In the revised version, this is now Section 2.3.  We have added a paragraph in blue with additional information.
>
> >Figure 3 compares Swin-T and Chromaformer-T on the 12-class HSV dataset. Several questions arise:
> >>    Reporting overall accuracy alone would be sufficient, .... achieves faster convergence.
>
> Answer: Showing that per-class performance is maintained or exceeded is important in remote sensing.  We strongly prefer to keep this more fine-grained evaluation.
>
> >>    Only the tiny variants are compared, despite the fact ....architectures would strengthen the analysis.
>
> Answer: We added the comparison between swin-b and chromaformer-b(see supplementary material A.6), we believe that the comparison is informative since the parameter count is large and performance of Chromaformer is consistently better than Swin across tiny and big models.
>
> > Figures 5, 8, and 9 present qualitative results .... understand where and why the method struggles.
>
> Answer: We added the raw input image in figure 5,8 and 9 for better readability.
>
> >>    Additionally, there is no state-of-the-art (SOTA) qualitative comparison for Figures 8 and 9 on the Indian Pines and Pavia University datasets.
>
> Answer: We added the qualitative output comparison between models in figure 8 and 9.
>
> >>    In Figure 8, the prediction maps with error overlays ....Moreover, Figures 8 and 9 are not adequately discussed in the main text.
>
> Answer: The original error overlay is hard to read since there is only one mis-classified pixel for chromarformer, we substituted Figure 8 with new version including Swin output. We updated captions of Figure 8 and 9 for clarification.
>
> >>    There is also no quantitative SOTA comparison for the Indian Pines and Pavia University datasets, which would be expected given their widespread use.
>
> Answer: Comparison is added for SOTA models in section 3.5.
>
> > The paper places disproportionate emphasis on the BVM dataset. ....are missing for the other two datasets.
>
> Answer: Confusion matrix of PaviaUni dataset was added in supplementary material A.7.
>
> > An ablation study of the SDM module is missing. For example, .... would strengthen the technical contribution.
>
> Answer: We include SDM ablations via (a) early vs. late placement and (b) replacement with SE/CBAM in Section A.5. SDM is not multi-head attention, so it has no independent number of heads hyperparameter, and embedding dimension is fixed by the backbone width, so it cannot be tuned independently without changing the model scale (t/s/b/l/h). More extensive SDM-specific variants (e.g., low-rank/sparse dependencies) are left for future work and are stated as a limitation in Section 4.

---

### Review · Reviewer_3a3w · 2025-12-24

**Summary Of Contributions:**

The paper introduces ChromaFormer, a family of scalable transformer-based architectures tailored for large-scale multi-spectral land cover classification. The key technical contribution is the Spectral Dependency Module (SDM), a lightweight attention mechanism that explicitly models inter-band dependencies by performing attention across spectral channels rather than spatial tokens. This module is integrated into a Swin Transformer backbone to enable joint spectral–spatial reasoning while retaining computational scalability.

The work is evaluated primarily on the Biological Valuation Map (BVM) of Flanders, a uniquely large (13,500 km^2), densely labeled, expert-curated dataset derived from Sentinel-2 imagery. The authors conduct a systematic scaling study across model sizes (27M to 655M parameters), demonstrating strong performance gains with increasing capacity. ChromaFormer consistently outperforms CNN baselines (ResNet, UNet++) and improves upon Swin Transformers at comparable scales, achieving up to 96.7% overall accuracy on BVM. Additional experiments on Indian Pines and Pavia University benchmarks indicate strong generalization to standard hyperspectral datasets.

Strengths
- Clear architectural motivation for explicit spectral modeling.
- Convincing large-scale empirical evaluation on an unusually dense and high-quality dataset.
- Careful analysis of scaling behavior across architectures and model sizes.
- The SDM is simple, interpretable, and easily portable to other backbones.

Weaknesses
- Limited geographic diversity in the primary large-scale evaluation (BVM is region-specific).
- The novelty of SDM is incremental relative to prior channel-attention mechanisms, and its theoretical properties are not deeply analyzed.
- Comparisons against recent remote sensing foundation models or large pretrained transformers are limited.

**Additional Comments:**

Overall, this is a strong empirical contribution that combines architectural clarity with unusually large-scale experimentation for the remote sensing domain. While the methodological novelty is incremental rather than disruptive, the rigor of the evaluation and the clarity of the scaling analysis make the paper a good fit for TMLR. With improved contextualization of SDM relative to prior work and a slightly broader generalization discussion, the paper would be well-positioned for acceptance.

**Audience:**

Yes

**Audience Explanation:**

The paper is likely to interest multiple segments of the TMLR audience, particularly researchers working on:

- Representation learning and scaling laws beyond natural image benchmarks,
- Transformers for structured, high-dimensional inputs (e.g., spectral, temporal, or scientific data),
- Applied machine learning for earth observation and environmental monitoring.

Beyond its domain-specific relevance, the work provides a concrete case study on how architectural inductive biases (here, spectral attention) interact with scale, data density, and task structure. The large-scale evaluation and careful scaling analysis make the findings informative even for readers not directly focused on remote sensing.

**Broader Impact Concerns:**

The work focuses on land cover classification using publicly available satellite imagery and does not introduce direct ethical risks.

**Claims And Evidence:**

Yes

**Claims Explanation:**

The paper’s central claims that explicit spectral dependency modeling improves multi-spectral classification performance and that transformer-based architectures scale more effectively than CNNs for dense remote sensing tasks are well supported by empirical evidence.

The authors provide:
- A controlled architectural comparison between Swin Transformers with and without SDM, isolating the contribution of spectral attention.
- Extensive scaling experiments across multiple orders of magnitude in parameter count, showing consistent accuracy gains and favorable scaling coefficients.
- Per-class accuracy analyses and confusion matrices that substantiate improvements on spectrally complex and minority classes.
- Cross-dataset validation on Indian Pines and Pavia University, demonstrating that gains are not confined to the BVM dataset.

While the work is largely empirical, the evidence presented is thorough, clearly reported, and aligned with the claims made. The absence of deeper theoretical analysis does not undermine correctness, but rather limits interpretability.

**Requested Changes:**

Critical:
- Stronger positioning relative to prior spectral attention mechanisms: The SDM is conceptually related to channel-attention modules (e.g., SE, CBAM) and prior spectral transformers. A more explicit comparison, both conceptual and empirical, would clarify what is fundamentally new versus adapted.

- Broader generalization analysis: Since the main large-scale results rely on a single geographic region, evaluation on at least one additional large-area dataset (or a clear discussion of expected domain shift) would strengthen the generality of the conclusions.

Non-critical:
- Ablation on SDM complexity and placement beyond early vs. late fusion, including parameter count or attention-head sensitivity.
- Discussion of computational trade-offs, especially memory and training stability at the largest scales.
- Comparison with recent remote sensing foundation models or large pretrained transformers, even if limited to smaller-scale experiments.

---

> ### Author Response · Authors · 2026-02-19
>
> Dear reviewer,
>
> Thanks for your suggestions, we carefully read through your comments and made adjustments accordingly. Below are answers to some of your requested changes. Main text related to these contents is marked in olive color.
>
> >Limited geographic diversity in the primary large-scale evaluation (BVM is region-specific).
>
> Answer: Although the main benchmarking is on the BVM, we have provided evaluation also on other public benchmarks.  The architectural developments are in no way specific to a geographic region and represent generic operations over spectra.  We have clearly stated the geographic limitations in the last paragraph of Section 4.  A comparison on LandCoverNet is included in Figure 12.
>
> > The novelty of SDM is incremental relative to prior channel-attention mechanisms, and its theoretical properties are not deeply analyzed.
>
> Answer: Novelty is not a criterion for review in TMLR.  Please see https://jmlr.org/tmlr/acceptance-criteria.html for review criteria.  We have not made unsubstantiated theoretical claims in the paper.
>
> > Comparisons against recent remote sensing foundation models or large pretrained transformers are limited.
>
> Answer: We have outlined this in the limitations section (Section 4, last paragraph).
>
> > Stronger positioning relative to prior spectral attention mechanisms: The SDM is conceptually related to channel-attention modules (e.g., SE, CBAM) and prior spectral transformers. A more explicit comparison, both conceptual and empirical, would clarify what is fundamentally new versus adapted.
>
> Answer: We have added a discussion in Section A.4.
>
> > Broader generalization analysis: Since the main large-scale results rely on a single geographic region, evaluation on at least one additional large-area dataset (or a clear discussion of expected domain shift) would strengthen the generality of the conclusions.
>
> Answer: A comparison on LandCoverNet (North America) is included in Figure 12.
>
> > Discussion of computational trade-offs, especially memory and training stability at the largest scales.
>
> Answer: At the largest scales, ChromaFormer exhibits significant computational trade-offs, particularly in terms of memory usage and training stability. As model size increases (e.g., ChromaFormer-h with 656 million parameters), memory consumption grows accordingly, with each 40GB NVIDIA A100 GPU using up to 35GB of memory. This scaling demands careful management of batch sizes and hyperparameters tuning to ensure the model fits within available GPU memory. While larger models achieve higher accuracy, they also require more memory and computational power, increasing training time per epoch.
>
> > Comparison with recent remote sensing foundation models or large pretrained transformers, even if limited to smaller-scale experiments.
>
> We have noted this as a limitation in the last paragraph of Section 4.

---

### Review · Reviewer_6jWn · 2026-02-05

**Summary Of Contributions:**

The paper introduces ChromaFormer: a modified version of the famous transformer architecture that powers a lot of modern AI. Apart from this, the paper has two major contributions:
* It introduces the Spectral Dependency Module which shares the central idea of an attention layer (possibly just in different spectral domains) to compute attentions between different spectral representations.
* It shows that transformer scaling is still as effective even with the spectral dependency module (scaling efficiency is on equal footing to the Swin Transformer, with a slight increase in accuracy for the ChromaFormer) -- this is extremely important for learning complex data patterns with large transformers.

**Additional Comments:**

No additional comments.

**Audience:**

Yes

**Audience Explanation:**

Beyond the hyper spectral imaging community, this paper would be interesting to a large class of audience that are interested in using transformer architecture embedded/modified with a specific additional component designed for their use case. This paper has positive results (for the specific case of hyper spectral imaging) that the scaling efficiency still holds, like in other generic successful transformers.

**Broader Impact Concerns:**

N/A.

**Claims And Evidence:**

Yes

**Claims Explanation:**

Yes, the claims made are supported by accurate, convincing and clear evidence.

1. They choose a well-suited dataset.
2. They show scaling performance.
3. They show accuracy measures.


What is (possibly) not clear is the use of the validation dataset.

**Requested Changes:**

1. I am assuming the Overall accuracy and Per class classification accuracy were calculated on the test set (a clarifying statement in the paper would help). What is the validation set being used for?
2. The subsection, 2.2 Transformer backbone, is very unclear. I would suggest making it clear by adding information that can help the reader reproduce the architecture by themselves. A diagram of where the spectral dependency module goes into the transformer backbone would be very helpful (a diagram of the entire architectural design would be very helpful) : some idea can be gained by the early fusion and late fusion discussion in the appendix, but still a diagram seems necessary. An example would be something like Figures 3 (a) and (b) of Swin transformer paper (https://arxiv.org/pdf/2103.14030).
3. Projected embedding length in Figure 2 is not mentioned elsewhere: do you refer to the number of embedding channels that you mention later? Please clarify this.
4. Related to the number of embedding channels, the dataset you use has 13 channels, does the number of embedding channels represent a dimension where these 13 channels are projected (expanded) to: that is does it mean that you increase the dimension 13 to a higher number (like 96)? Please clarify this.
5. This sentence in 2.2 : Each stage consists of multiple Swin Transformer blocks with increasing receptive field and hidden dimensions, needs more explanation as I do not (at least in a straightforward manner) understand the idea of receptive field in transformers.

---

> ### Author Response · Authors · 2026-02-19
>
> Dear reviewer,
>
> Thanks for your time and contribution, below is the answers to your requested changes, main text related to these points are marked in blue.
>
> > 1. I am assuming the Overall accuracy and Per class classification accuracy were calculated on the test set (a clarifying statement in the paper would help). What is the validation set being used for?
>
> Answer: We added an explanation in Section 2.4.  The validation set was used for hyperparameter tuning and early stopping, while the test set is used for final performance metrics.
>
> > 2. The subsection, 2.2 Transformer backbone, is very unclear. I would suggest making it clear by adding information that can help the reader reproduce the architecture by themselves. A diagram of where the spectral dependency module goes into the transformer backbone would be very helpful (a diagram of the entire architectural design would be very helpful) : some idea can be gained by the early fusion and late fusion discussion in the appendix, but still a diagram seems necessary. An example would be something like Figures 3 (a) and (b) of Swin transformer paper (https://arxiv.org/pdf/2103.14030).
>
> Answer: A newer version of figure 2 and Section 2.2 is added in the revised manuscript.  As mentioned at the end of Section 2.3, code will also be made available facilitating reproduction of results.
>
> > 3. Projected embedding length in Figure 2 is not mentioned elsewhere: do you refer to the number of embedding channels that you mention later? Please clarify this.
>
> Answer: Detailed explanation is added in Section 2.3.  A dimension of C=96 was used in experiments.
>
> > 4. Related to the number of embedding channels, the dataset you use has 13 channels, does the number of embedding channels represent a dimension where these 13 channels are projected (expanded) to: that is does it mean that you increase the dimension 13 to a higher number (like 96)? Please clarify this.
>
> Answer: Detailed explanation is added in Section 2.3 (see above).
>
> > 5. This sentence in 2.2 : Each stage consists of multiple Swin Transformer blocks with increasing receptive field and hidden dimensions, needs more explanation as I do not (at least in a straightforward manner) understand the idea of receptive field in transformers.
>
> Answer: An explanation is added in Section 2.3 starting with the text "Similar to CNNs, the hierarchical nature of Swin-like transformers merges image patches at deeper layers."

---

### Decision · Action_Editor_viBg · 2026-04-08

**Recommendation:** Accept as is

**Audience:**

Yes

**Audience Explanation:**

The main scope of the paper is quite applied: It proposed a method for a specific classification task in remote sensing. The technical solution, however, is a novel machine learning artefact, a new deep learning architecture that is well justified and sufficiently connected to the literature. The architecture is detailed enough to warrant evaluation and publication in a machine learning venue, even though the paper  would be suitable also for a remote sensing venue.

The paper will be of interest for a broad range of applied machine learning researchers working in remote sensing. Outside of remote sensing, it is a good example of developing and evaluating a scalable task-specific transformer architecture and hence the work contributes to the general literature on transformers. Moreover, the Spectral Dependency Module may be relevant for multi-spectral data sources also outside remote sensing.

**Claims And Evidence:**

Yes

**Claims Explanation:**

The paper proposes a transformer-based model designed for land cover classification from satellite images. The main claim is good accuracy, which is properly validated with an empirical comparison on a large-scale evaluation task where the proposed method achieves substantially higher accuracy compared to a relatively strong baseline. The evaluation does not consider broad range of alternative models, but is still sufficient given the near-perfect accuracy of the proposed method on a recent benchmark task. Additional claims relating to scaling are evaluated with appropriate controlled studies.